# Intrinsic Disorder in BAP1 and Its Association with Uveal Melanoma

**DOI:** 10.3390/genes13101703

**Published:** 2022-09-22

**Authors:** Mak B. Djulbegovic, David J. Taylor, Vladimir N. Uversky, Anat Galor, Carol L. Shields, Carol L. Karp

**Affiliations:** 1Bascom Palmer Eye Institute, University of Miami, Miami, FL 33136, USA; 2Department of Molecular Medicine and USF Health Byrd Alzheimer’s Research Institute, Morsani College of Medicine, University of South Florida, Tampa, FL 33613, USA; 3Ophthalmology, Miami Veterans Affairs Medical Center, Miami, FL 33136, USA; 4Research Services, Miami Veterans Affairs Medical Center, Miami, FL 33136, USA; 5Ocular Oncology Service, Wills Eye Hospital, Thomas Jefferson University, Philadelphia, PA 19107, USA

**Keywords:** uveal melanoma, BAP1, intrinsically disordered protein (IDP), intrinsically disordered protein regions (IDPR), protein–protein interaction network

## Abstract

Background: Specific subvariants of uveal melanoma (UM) are associated with increased rates of metastasis compared to other subvariants. BRCA1 (BReast CAncer gene 1)-associated protein-1 (BAP1) is encoded by a gene that has been linked to aggressive behavior in UM. Methods: We evaluated BAP1 for the presence of intrinsically disordered protein regions (IDPRs) and its protein–protein interactions (PPI). We evaluated specific sequence-based features of the BAP1 protein using a set of bioinformatic databases, predictors, and algorithms. Results: We show that BAP1’s structure contains extensive IDPRs as it is highly enriched in proline residues (the most disordered amino acid; *p*-value < 0.05), the average percent of predicted disordered residues (PPDR) was 57.34%, and contains 9 disorder-based binding sites (ie. molecular recognition features (MoRFs)). BAP1’s intrinsic disorder allows it to engage in a complex PPI network with at least 49 partners (*p*-value < 1.0 × 10^−16^). Conclusion: These findings show that BAP1 contains IDPRs and an intricate PPI network. Mutations in UM that are associated with the BAP1 gene may alter the function of the IDPRs embedded into its structure. These findings develop the understanding of UM and may provide a target for potential novel therapies to treat this aggressive neoplasm.

## 1. Introduction

Uveal melanoma (UM) is the most common primary intraocular malignancy and has an age-adjusted incidence of 5.1 per million [1,2]. The neoplasm arises from the melanocytes in different parts of the uvea, specifically the choroid, ciliary body, or iris. UM comprises 85% of melanomas arising in the eye, with the remainder of eye-related melanomas arising mostly from the conjunctiva [2]. Most patients present between the ages of 50 years to 70 years and risk factors that have been associated with UM include fair skin, light iris color, congenital ocular melanocytosis, melanocytoma, and BAP1 cancer predisposition syndrome [3,4,5,6]. 

UM may present without any symptoms and may be discovered incidentally during a routine eye examination. However, approximately >70% of patients may have minor visual disturbances, such as flashes, floaters, or other visual field defects [4]. The diagnosis is based upon a comprehensive ophthalmic examination, which is followed by further characterization with specialized noninvasive testing methods, such as fundus photography, ultrasonography, ultrasound biomicroscopy, optical coherence tomography, autofluorescence, and fluorescein angiography [5,7]. Although these non-invasive methods can be accurately used in many UM cases, a biopsy of the lesion through fine-needle aspiration may be necessary to yield a definitive diagnosis [4,8].

When treating UM, the primary goal is complete tumor control for prevention of metastatic spread and the secondary goal is the preservation of vision if possible. Historically, enucleation was the mainstay of treatment, as this would provide confirmatory histopathological diagnosis and help guide further decisions. In the 1970s, the treatment paradigm slowly shifted towards using Iodine-125 (125I) brachytherapy, which is a form of radiation therapy applied directly over the tumor on the episcleral surface [9,10]. The Collaborative Ocular Melanoma Study (COMS) showed that there was no significant difference in mortality when comparing 125I brachytherapy to enucleation [11]. This landmark study now defines how UM is managed.

Today, the American Brachytherapy Society has standardized the use of brachytherapy to treat UM [12]. Other treatments that are being compared to brachytherapy include proton beam radiotherapy [13], and smaller tumors may be treated with transpupillary thermotherapy, photodynamic therapy, photocoagulation, and now investigations into nanoparticle therapy [14,15,16,17]. Although local treatment for primary UM has prevented local recurrence in over 95% of cases, up to 50% of patients develop hematogenous metastases that are believed to be micrometastasis even before therapy [5,18,19]. 

Given the high rates of metastasis following local tumor control, there is a need to further detail the factors that are driving the aggressive behavior of UM. The rise in precision and genetic medicine has provided motivation to elucidate the molecular basis of disease, especially malignancies, including UM. The genomic landscape of UM, characterized by a small number of highly recurrent mutations, has been studied and profiled via gene expression profiling (GEP), which has been used as a prognostic tool to predict metastatic risk [20,21]. GEP can characterize UM tumors in two subgroups: Class 1, which confers a low metastatic risk, and Class 2, which confers a high metastatic risk [20,21]. Class 2 tumors contain a cluster of “progression mutations” in SF3B1, EIF1AX, and BAP1 [20,21]. Of these three genes, BAP1 mutations and loss of function have consistently been linked to aggressive sub-variants of UM [20,21]. 

The BRCA1 (BReast CAncer gene 1)-associated protein (BAP1) gene encodes a 729-residue-long tumor suppressor protein that functions via multiple incompletely understood pathways. BAP1 is located on chromosome 3p21.1, which is frequently lost or mutated in uveal melanoma [21,22,23]. Mutations in BAP1 consist of missense and nonsense mutations, frame-shift deletions, non-frame-shift deletion, and four splice site mutations [23]. These discoveries and associations are key to better understanding UM; however, additional work is needed to further comprehend how these genetic mutations alter various functions of this important protein. Specifically, it is unclear how these mutations impact the functional proteomics of BAP1 and its interaction with other proteins. 

Many aspects of UM’s proteome are likely impacted by genetic mutations associated with the neoplasm. In the interest of our group, we aim to understand UM in the context of the protein intrinsic disorder phenomenon, focusing on quantifying the intrinsically disordered protein regions (IDPRs) in BAP1 and characterizing its protein–protein interaction (PPI) network. IDPRs and intrinsically disordered proteins (IDPs) have been linked to neoplastic development [24,25,26,27]. IDPs and proteins with IDPRs are commonly found in proteomes of all living organisms and viruses [28,29,30,31]. These functional proteins or protein regions are characterized by the lack of stable 3D structure and possess highly flexible structures containing multiple functional segments. IDPRs are likely to be involved in an array of diverse biological functions [32,33,34]. Our group’s previous studies have demonstrated associations between IDPRs and the development of ocular surface squamous neoplasia and conjunctival melanoma [35,36,37]. Importantly, in a shift from structure-specific drug discovery campaigns, IDPRs have begun to be targeted to treat various cancers [38,39,40]. We posit that, like many other cancer-related proteins, BAP1 has extensive intrinsic structural disorder (i.e., IDPRs). If BAP1 contains high levels of intrinsic structural disorder, then these molecular features should be considered as potentially viable targets for future therapies directed at UM.

## 2. Materials and Methods

An initial literature search with the key terms “uveal melanoma” and “BAP1” was done to understand the current treatment landscape of UM and to identify genetic mutations associated with UM. Articles used in the literature search were collected from the National Library of Medicine MEDLINE databases. All forms of published scientific articles were considered, including original research, meta-analyses, and systematic reviews. All articles were limited to publications in the English language. Through our literature search, we identified 66 unique BAP1 mutations reported by Ewens et al., Decatur et al., and Harbour et al. [21,41,42]. A disorder-based proteomics analysis was also performed to quantify the levels of intrinsic disorder (i.e., IDPRs) within BAP1 and to assess the complexity of its PPI network. 

### 2.1. Protein Sequences

The Universal Protein Resource (UniProt; available at: https://uniprot.org; accessed on 15 June 2022) is a database that houses comprehensive protein information [43]. The search terms used were “gene: BAP1” and “Homo sapiens (Human)”. The canonical amino acid sequence in texted-based format (FASTA) was selected from the UniProt entry (UniProt ID: Q92560). 

### 2.2. Structural Assessment

The UniProt entry used above (UniProt ID: Q92560) also contained structural information. We selected the “AlphaFoldDB” entry as AlphaFold is the most accurate and accepted computational method to predict structural aspects of proteins [44,45].

### 2.3. Quantification of Intrinsic Disorder

The BAP1 sequence was subjected to analysis using the Composition Profiler (available at: http://www.cprofiler.org/; accessed on 15 June 2022) [46]. The profiler allowed for the analysis of enrichment or depletion of common amino acids in BAP1. The content of each amino acid in the query protein is compared to background set of proteins that is comprised of highly structured and ordered proteins, the PDB Select 25 [46]. The normalized enrichment or depletion is evaluated as (Cx–Corder)/Corder, where Cx is the content of a given residue in its query protein (i.e., BAP1) and Corder is the content of the same residue in the PDB Select 25.

The BAP1 sequence was also analyzed using a set of six commonly used per-residue disorder predictors: PONDR^®^ VLXT [47], PONDR^®^ VL3 [48], PONDR^®^ VLS2 [49,50], PONDR^®^ FIT [51], and IUPred2 (Short) and IUPred2 (Long) [52,53]. The corresponding data were assembled using an in-house web crawler that aggregates the outputs of these six well-known disorder predictors and also provides mean disorder predictions for query proteins by averaging the outputs of these six predictors. We also used PONDR^®^ VLS2 results to quantify local intrinsic disorder predisposition for each BAP1 mutations that we identified in the literature.

Potential disorder-related functionality of BAP1 was analyzed using the Database of Disordered Protein Prediction (D2P2) database (http://d2p2.pro/; accessed on 15 June 2022) [54] that utilizes the outputs of IUPred [52], PONDR^®^ VLXT [47], PrDOS [55], PONDR^®^ VSL2B [49,50], PV2 [54], and ESpritz [56] to show disorder predispositions of a query protein and to evaluate agreement between these predictors. The platform also shows the positions of functional SCOP (Structural Classification of Proteins) domains [57,58] predicted by the SUPERFAMILY predictor [59]. The D2P2 output also includes information identified by the ANCHOR algorithm, which predicts disorder-based binding sites that are disordered and expected to fold into ordered structures when they interact with specific partners (i.e., molecular recognition features (MoRFs)) [60]. The D2P2 platform also predicts possible posttranslational modifications (PTMs) using the outputs of the PhosphoSitePlus [61]. Lastly, we analyzed how known UM-related mutations might affect local disorder propensity of BAP1. 

### 2.4. Protein–Protein Interaction Network

To assess the potential functional implications of intrinsic disorder in BAP1, we subjected the amino acid sequence to The Search Tool for the Retrieval of Interacting Genes (STRING) (URL: https://string-db.org/, accessed on 15 June 2022). STRING is a quality-controlled database that uses experimentally and computationally derived data to detail functional interactions between proteins [62]. The STRING search included the canonical BAP1. The settings for the minimum required interaction score were set to the highest confidence (0.900), and the maximum number of interactions to show was assigned to the highest possible value of 500.

## 3. Results

This section may be divided by subheadings. It should provide a concise and precise description of the experimental results, their interpretation, as well as the experimental conclusions that can be drawn.

### 3.1. Protein Sequence

The canonical amino acid sequence of the BAP1 protein in FASTA format (Appendix A) was used for our bioinformatics analysis. BAP1 is a deubiquitinating enzyme from the ubiquitin C-terminal hydrolase (UCH) family that plays a key role in the chromatin remodeling by mediating deubiquitylation of histone H2A and HCFC1 (Host Cell Factor C1) [63,64,65,66]). The protein sequence consists of 729 amino acids and includes several functional domains and regions. The domains include (1) a cysteinase domain (or ubiquitin carboxyl-terminal hydrolase, family 1, residues 1–236), (2) an HBM-like motif (for host cell factor-1 (HCF-1)-binding motif, NHNY, located within a short sequence (residues 356–385) that is highly conserved among species [67]), (3) a C-terminal BRCA1 interacting domain (residues 598–729), which includes a ubiquitin carboxyl-terminal hydrolase UCH37-like domain (ULD) at residues 675–693 and two nuclear localization sequences (residues 656–661 and 717–722).

### 3.2. Structural Assessment

The modeled 3D structure of BAP1 modeled by AlphaFold (Figure 1) demonstrates that the protein has an ordered domain surrounded by long regions of structural disorder. Within BAP1 structure (see Figure 1), these intrinsically disordered regions appear as string-like or spaghetti-like segments. Importantly, although these IDPRs do not have definitive structures, they are likely to be involved in an array of diverse biological functions [32,33,34].

### 3.3. Quantification of Intrinsic Disorder

The next phase of our disorder-based computational analysis turned toward quantifying the IDPRs identified in our structural assessment. We first constructed a normalized composition profile of the amino acids that are part of the BAP1 primary amino acid sequence (Figure 2). In the corresponding plot, amino acids are arranged from most order-promoting residues (i.e., cysteine (C), tryptophan (W), isoleucine (I), tyrosine (Y), phenylalanine (F), leucine (L), histidine (H), valine (V), asparagine (N), and methionine (M)) to most disorder-promoting residues (i.e., arginine (R), threonine (T), aspartate (D), glycine (G), alanine (A), lysine (K), glutamine (Q), serine (S), glutamate (E), and proline (P)). The positive and negative values produced in the composition profile analysis indicated enrichment or depletion of the indicated residue, respectively. Half (5 of 10) disorder-promoting residues showed enrichment (R, Q, S, E, and P) and only 3 of 10 revealed enrichment of order-promoting residues (L, H, and V). The most disorder-promoting amino acid, proline, had the highest relative enrichment of all the amino acids in this protein (*p*-value < 0.05). In fact, prolines account for 7.5% of all residues in BAP1, which is significantly higher than the proline content averaged over the entire UniProt database (4.7%). In addition, BAP1 has a significant proportion of its sequence comprised of other disorder-promoting residues. These compositional biases indicate that BAP1 has the amino acid composition that promotes elements of disorder in its structure. 

After establishing the amino acid composition propensity of BAP1 for the disorder, we aimed to quantify the intrinsic disorder in this protein. To this end, we utilized six commonly used per-residue predictors of intrinsic disorder to quantify and visualize the IDPRs in BAP1 (Figure 3A). The tools used in this portion of the study were PONDR^®^ VLXT, PONDR^®^ VSL2, PONDR^®^ VL3, PONDR^®^ FIT, IUPred short, and IUPred long. These tools allow for the categorization based on the established criteria for the classification of proteins based on their PPDR scores, where proteins are considered highly ordered (PPDR < 10%), moderately disordered (10% ≤ PPDR < 30%), and highly disordered (PPDR ≥ 30%) [68]. These tools indicated that BAP1 is characterized by a high percentage of predicted protein disordered residues (PPDR), which were 53.36% (PONDR^®^ VLXT), 61.04% (PONDR^®^ VSL2), 57.61% (PONDR^®^ VL3), 50.34% (PONDR^®^ FIT), 43.07% (IUPred short), and 50.21% (IUPred long). In this analysis, the average PPDR of BAP1 protein is 52.6 ± 4.6%, which classifies it as highly disordered. These multiparametric disorder evaluation results are in line with the outputs of the structural analysis of human BAP1 by AlphaFold.

From our literature review, we were able to identify a list of mutations that were associated with the development of UM. We analyzed the local disorder predispositions of sites affected by these mutations. Results of this analysis are summarized in Appendix A, which includes 66 entries. Based on the analysis of 54 UM samples, Ewens et al. reported 52 unique BAP1 mutations that were defined as pathogenic or likely pathogenic, with 43 tumors carrying somatic mutations, and remaining 11 tumors carrying germline mutations [42]. Among those mutations, 15 were splice-site mutations with unknown effects on protein sequence, whereas mutations with known effects on protein sequence include seven missense mutations, three in-frame deletions, and 29 mutations causing premature truncation [42]. We complemented these data by the results on four missense, six truncating, and two in-frame deletions reported by Decatur et al. [41], and also included data on four missense, seven truncating, and four in-frame deletions from Harbour et al. [21]. Appendix A shows that on average, 66 UM-related mutations in BAP1 are characterized by the disorder score of 0.48 ± 0.31 PONDR^®^ VSL2 (i.e., high disordered by PPDR), indicating that many of these mutations affected intrinsically disordered regions. In fact, 28 mutations happened within highly disordered regions (i.e., regions with the disorder score exceeding 0.5), 26 mutations were in flexible regions (i.e., regions with the disorder score ranging from 0.15 to 0.5), and only 12 mutations were in regions predicted as mostly ordered (i.e., regions with disorder scores below 0.15). 

To shed additional light on the potential role of intrinsic disorder in functionality of BAP1, the D2P2 platform was used (Figure 3B). The D2P2 platform output contains a corresponding functional disorder profile for BAP1 and shows that in addition to the presence of high levels of intrinsic disorder, BAP1 is predicted to have 9 MoRFs (residues 258–265, 289–306, 309–326, 330–401, 410–479, 497–508, 526–550, 555–575, and 673–684) that cover 35.1% of it sequence (or almost 70% of its disordered residues). Furthermore, BAP1 has multiple phosphorylation sites, two ubiquitination sites and 3 methylation sites, all located within IDPRs of this protein. BAP1 has a strong tendency for its IDPRs to be post translationally modified and is in agreement with the well-established fact that disordered regions of proteins often serve as primary targets for various post-translational modifications (PTMs) [69,70,71]. Therefore, BAP1 is highly disordered and can use its intrinsic disorder for interaction with target proteins, and these interactions are likely to be fine-tuned via multiple PTMs.

### 3.4. Protein–Protein Interaction Network

The bioinformatics analyses demonstrated that intrinsic disorder is present in BAP1. We aimed to further understand the potential functional implications of structural properties. Our STRING analysis allowed us to assess the binding capabilities of BAP1 (Figure 4). This analysis, which was conducted using the highest confidence of 0.9 for the minimum required interaction score, revealed that BAP1 can interact with 49 other proteins in its 1st shell interaction network. The number of predicted protein–protein interactions in the BAP1 protein interaction network is 599, and the expected number of edges for randomly selected proteins of the same-sized network is 101 (*p*-value < 1.0 × 10^−16^).

## 4. Discussion

To the best of our knowledge, this work represents the first study analyzing the presence of intrinsic disorder in the BAP1 protein which has mutations associated with uveal melanoma. Our proteomics analysis confirmed the presence of intrinsically disordered protein regions (IDPRs) throughout the BAP1 protein. Furthermore, this protein is shown to contain elements of structural disorder, as demonstrated by the spaghetti-like entities visualized with AlphaFold. In addition, proline, the most disordered amino acid, was the most enriched amino acid in BAP1 (*p*-value < 0.05). Our subsequent bioinformatics analysis revealed that BAP1 is characterized by an average PPDR of 52.6 ± 4.6% and has nine molecular recognition features (MoRFs) that cover 35.1% of its amino acid sequence, including almost 70% of BAP1 disordered residues. In addition, our study shows that BAP1 has multiple sites of various PTMs and is characterized by high binding promiscuity as it is engaged in numerous protein–protein interactions. These constellations of findings support our hypothesis that intrinsic disorder is an important functional feature of BAP1. Knowing that intrinsically disordered proteins may be associated with the development of neoplasia, these findings are important for considering the pathogenesis of aggressive subtypes of UM. 

Unfortunately, the prognosis is poor for patients diagnosed with advanced or aggressive UM with the BAP1 mutation. There is an obvious need to develop novel treatments to manage UM. Our analysis showed that the BAP1 protein demonstrates high propensity for disorder throughout its structure. These disordered regions provide the protein with dynamic binding capabilities that allow the creation of a highly complex protein–protein interaction network. IDPRs are fluid structures, and their interface extends therapeutic options beyond those previously considered in the traditional structure–function paradigms that direct most modern drug discovery campaigns [39,72]. BAP1 also functions as a tumor suppressor gene, which are increasingly targeted for potential therapeutic benefit [73,74]. In light of our findings and in conjunction with recent efforts to target tumor suppressor genes, future attempts to target molecular features of UM should consider IDPRs in BAP1. Efforts to target these dynamic, structureless regions of BAP1 may identify solutions not otherwise considered with traditional structure-focused drug campaigns.

Mutations that impact the intrinsic disorder protein regions (IDPRs) have been implicated in developing other neoplasms, including conjunctival melanoma and ocular surface squamous neoplasia [35,36,37]. Dysregulation of disordered regions that leads to dysfunction of BAP1 may have a role in the development of UM. In line with these considerations, our analysis of 66 UM-related mutations revealed that they are characterized by the intrinsic disorder score of 0.48 ± 0.31 as measured by PONDR^®^ VSL2, indicating that many of these mutations are likely to alter the function of intrinsically disordered regions. We also show that 28 UM-related mutations happened within highly disordered regions of BAP1, 26 mutations were within moderately disordered regions (i.e., flexible regions), and only 12 mutations (i.e., 18% of all UM-related mutations analyzed in this study) were within the mostly ordered regions (Appendix A). 

The mutational landscape of UM is diverse; therefore, there are likely many novel targets to consider for the management of UM. In the interest of our study, the IDPRs in BAP1 should be considered. There is evidence that BAP1 mutated UM tumors can be targeted through poly [ADP-ribose] polymerase (PARP) inhibition, albeit indirectly and in BAP1 loss tumor cell lines [75]. Other studies have shown that UM contains BAP1 with missense, nonsense, frame-shift deletions, non-frame-shift deletion, and four splice site mutations [23]. Another study demonstrated that the ubiquitin carboxyl hydrolase (UCH) domain in BAP1 is a hotspot region for mutations associated with UM, which lead to the cytoplasmic sequestration and loss of the protein [76]. Our findings suggest that these previously characterized mutations may lead to the loss or change of function in BAP1 protein domains and may lead to aberrant regulation of IDPRs. These alterations in IDPRs of BAP1 may lead to unique DNA methylation profiles as suggested by TCGA [77].

As with all studies, our study has limitations. The databases and tools utilized in this study do not provide a comprehensive nor exhaustive representation of the UM-related proteomics. Our work is a bioinformatics-based investigation, and translation to basic science and clinical outcomes are needed. We did not conduct an in-depth analysis of where specific mutations are found in BAP1 (i.e., within protein domains or IDPRs). A future study could provide an additional understanding of the impacts of these mutations on BAP1 structure, conformational dynamics, and function. Additional bench work will be needed to determine the role of IDPRs in BAP1. This work may lead to novel targets to treat specific subtypes of UM. Unfortunately, targeting IDPs and IDPRs by small molecules remains a challenge and strategies to develop protein disorder-based therapeutics are limited [36]. 

In addition to the above limitations, there are other genetic factors that need to be further studied. We did not include an analysis of other genetic mutations associated with UM that TCGA has identified, such as GNAQ (Guanine nucleotide-binding protein G(q) subunit α), GNA11 (Guanine nucleotide-binding protein subunit α-11), EIF1AX (Eukaryotic translation initiation factor 1A, X-chromosomal), SF3B1 (Splicing factor 3B subunit 1), CYSLTR2 (Cysteinyl leukotriene receptor 2), SRSF2 (Serine/arginine-rich splicing factor 2), MAPKAPK5 (MAP kinase-activated protein kinase 5), and PLCB4 (1-phosphatidylinositol 4,5-bisphosphate phosphodiesterase β-4) [77]. There is a strong likelihood that these proteins also display elements of intrinsic disorder, and additional work will be needed to confirm this hypothesis and detail the implications. Lastly, while BAP1 mutations are associated with aggressive subtypes of UM, in some cases, the loss of the entire chromosome 3 were reported (monosomy 3) [78,79,80,81]. Therefore, targeting IDPRs in BAP1 may not be a viable solution in these specific UM subvariants.

## 5. Conclusions

Our study demonstrates that intrinsic disorder in BAP1 may be associated with the development of uveal melanoma. Our group has previously detailed the associations between intrinsically disordered proteins and the development of ocular surface squamous neoplasia and conjunctival melanoma. Genetic changes associated with BAP1 have been linked to increased metastatic behavior in uveal melanoma. We found that BAP1 contains intrinsically disorder protein regions (i.e., average percent of predicted disordered residues (PPDR) was 57.34%) and a complex protein–protein interaction network (*p*-value < 1.0 × 10^−16^). Our findings suggest that mutations associated with BAP1 may alter the behavior of intrinsically disordered protein regions, potentially promoting neoplastic development. Better treatments for UM are needed, and we believe that novel targets should be explored and include intrinsically disordered protein regions in BAP1. 

## Figures and Tables

**Figure 1 genes-13-01703-f001:**
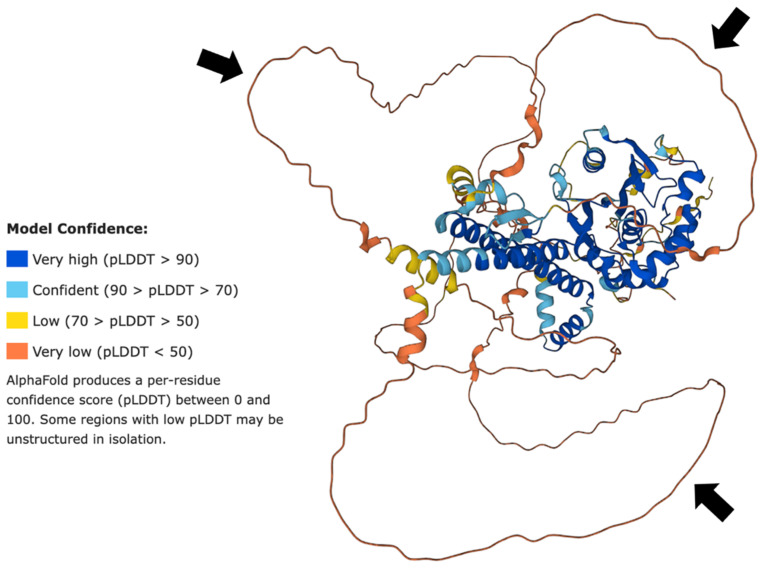
Structures generated for BAP1 by AlphaFold2. Regions with higher model confidence as measured by the predicted local distance difference test (pLDDT) typically represent α-helices. Regions with lower model levels represent intrinsically disordered protein regions. The black arrows indicate areas of high levels of disorder and lack of structure.

**Figure 2 genes-13-01703-f002:**
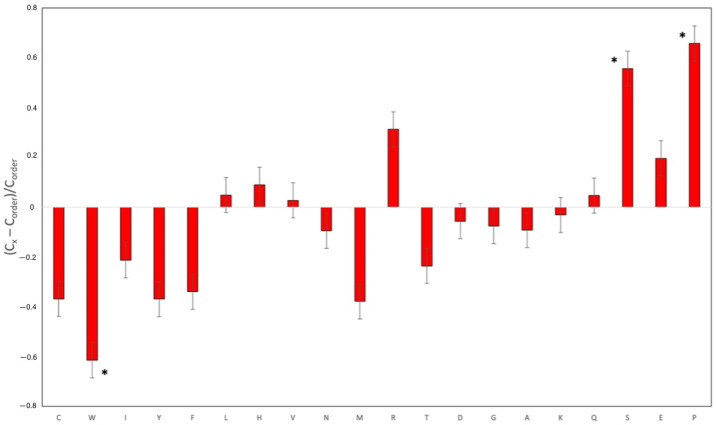
Amino acid composition profile of BAP1. The fractional difference is calculated as (Cx–Corder)/Corder, where Cx is the content of a given amino acid in the query set (BAP1), and Corder is the content of a given amino acid in the background set (Protein Databank Select 25). The amino acid residues are ranked from most order promoting residues ((i.e., cysteine (C), tryptophan (W), isoleucine (I), tyrosine (Y), phenylalanine (F), leucine (L), histidine (H), valine (V), asparagine (N), and methionine (M)) to most disorder promoting residues ((i.e., arginine (R), threonine (T), aspartate (D), glycine (G), alanine (A), lysine (K), glutamine (Q), serine (S), glutamate (E), and proline (P)). Positive values indicate enrichment, and negative values indicate depletion of amino acids. 7 of 10 order-promoting residues are depleted and 5 of 10 disorder-promoting residues are enriched. Residue bars marked with * are statistically significant for depletion (i.e., W) or enrichment (i.e., S and P) (*p*-value < 0.05)).

**Figure 3 genes-13-01703-f003:**
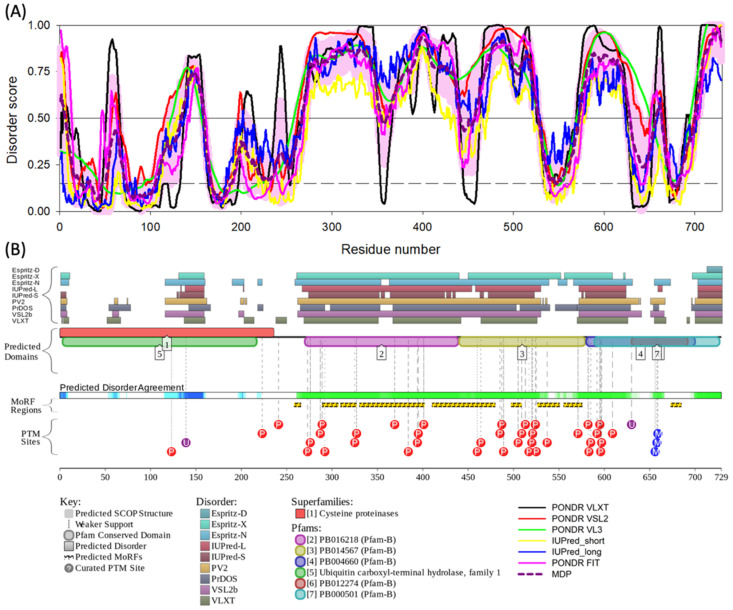
Evaluation of the intrinsic disorder predisposition of human BAP1. (**A**) Per-residue intrinsic disorder profiles generated by six commonly used intrinsic disorder predictors, PONDR^®^ VLXT, PONDR^®^ VL3, PONDR^®^ VSL2, PONDR^®^ FIT, IUPred short, and IUPred long. The thin black line at the disorder score of 0.5 is the threshold between order and disorder, where values above 0.5 are disordered residues and values below 0.5 are ordered residues. Thin black dashed line at disorder score 0.15 shows the threshold between flexible and mostly ordered region/residues. There are many segments of disorder seen throughout BAP1’s structure. In particular, the amino acid segment from 250 to 500 is highly enriched with intrinsic disorder. (**B**) The D2P2 platform output is used to assess the functional disorder profile for BAP1 protein. On the left of the figure, there are four identifiers for predictors used on the D2P2 platform. The top segment shows outputs from various per-residue disorder predictors, the second segment shows predicted protein domains, the third segment shows where molecular recognition features are located (MoRFs; i.e., disorder regions that become ordered when binding), and the fourth segment shows posttranslational modifications (PTM) sites. The per-residue predictors agree and align with the PONDR^®^ outputs from (**A**). The protein domains fall within regions that are predicted to be ordered or disordered. The MoRF regions are localized to regions of that demonstrate extensive disorder. The PTMs are also predominantly localized to areas of BAP1 that contain intrinsic disorder.

**Figure 4 genes-13-01703-f004:**
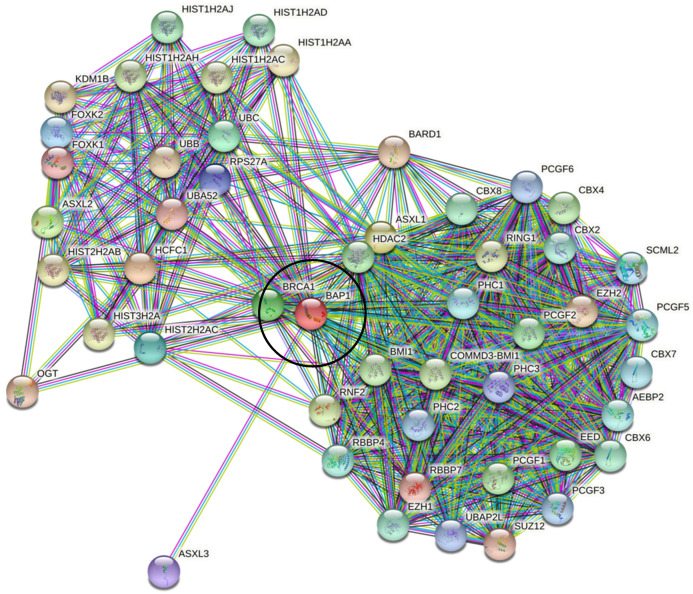
Search Tool for the Retrieval of Interacting Genes (STRING) output for BAP1. The graphic demonstrates that BAP1 (black circle) interacts with (lines) multiple proteins (colored circles). The ability for BAP1 to interact with 49 other proteins is likely possible through the utilization of IDPRs. The number of edges in this protein interaction network is 599, and the expected number of edges for randomly selected proteins of the same-sized network is 101 (*p*-value < 1.0 × 10^−16^).

## Data Availability

The data presented in this study are available on publicly accessible repositories. The data presented in this study are openly available in the Protein Data Bank at https://www.rcsb.org/ (accessed on 15 June 2022) and the Universal Protein Resource at https://uniprot.org (accessed on 15 June 2022). Protein structures and sequences were sourced from these databases and used in subsequent analyses.

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
