# Peer review of "Intrinsic Disorder in BAP1 and Its Association with Uveal Melanoma"

_genes, 2022, doi:10.3390/genes13101703_

Round 1
Reviewer 1 Report
It is a well written and planned study to describe the role of Intrinsic disorder protein region in BAP1 gene which is associated with UM.
Author Response
Thank you for your kind comment. Hopefully, the extensive intrinsic disorder analysis described in this study will improve the chances of success in drug discovery efforts, potentially yielding novel therapeutics to treat UM.

Reviewer 2 Report
This is a paper on the role of BAP1 in uveal melanoma.
The authors evaluated specific sequence-based features of the BAP1 protein using a set of bioinformatic databases, predictors, and algorithms. So, it is not own lab-work.
Still I do think that this paper contributes to the understanding of the role of BAP1 in uveal melanoma and I would advise to publish it.
Author Response
We appreciate the thoughtful feedback and your effort in reviewing the manuscript.
